Comparison of two molecular barcodes for the study of equine strongylid communities with amplicon sequencing

Courtot Élise 1
Boisseau Michel 1 2
Dhorne-Pollet Sophie 3
Serreau Delphine 1
Gesbert Amandine 4
Reigner Fabrice 4
http://orcid.org/0000-0001-7698-1253 Basiaga Marta 5
Kuzmina Tetiana 6 7
Lluch Jérôme 8
Annonay Gwenolah 8
http://orcid.org/0000-0001-8994-550X Kuchly Claire 8
Diekmann Irina 9
Krücken Jürgen 9
von Samson-Himmelstjerna Georg 9
Mach Nuria 2
Sallé Guillaume 1 guillaume.salle@inrae.fr
1 Animal Health, UMR1282 Infectiologie et Santé Publique, INRAE , Nouzilly , France
2 Animal Health, UMR1225 IHAP, Institut National de la Recherche pour l’Agriculture, l’Alimentation et l’Environnement (INRAE) , Toulouse , France
3 Université Paris-Saclay, INRAE, AgroParisTech , GABI, Jouy-en-Josas , France
4 Animal Physiology, UEPAO, Institut National de la Recherche pour l’Agriculture, l’Alimentation et l’Environnement (INRAE) , Nouzilly , France
5 University of Agriculture in Kraków , Kraków , Poland
6 Schmalhausen Institute of Zoology NAS of Ukraine , Kyiv , Ukraine
7 Institute of Parasitology, Slovak Academy of Sciences , Košice , Slovak Republic
8 GeT-PlaGe, Institut National de la Recherche pour l’Agriculture, l’Alimentation et l’Environnement (INRAE) , Toulouse , France
9 Institute for Parasitology and Tropical Veterinary Medicine, Freie Universität Berlin , Berlin , Germany
Gillespie Joseph
Electronic publication date: 2023 Apr 12
Publication date: 2023
Volume: 11
Electronic Location ID: e15124
Received 2022 Jun 16; Accepted 2023 Mar 3
Copyright: © 2023 Courtot et al.
Copyright year: 2023
Copyright holder: Courtot et al.
License: This is an open access article distributed under the terms of the Creative Commons Attribution License, which permits unrestricted use, distribution, reproduction and adaptation in any medium and for any purpose provided that it is properly attributed. For attribution, the original author(s), title, publication source (PeerJ) and either DOI or URL of the article must be cited.
License URL: https://creativecommons.org/licenses/by/4.0/

Keywords: Horse, Parasite, Nematode, Cyathostomin, Strongylid, Internal transcribed spacer 2, Cytochrome c oxidase subunit I, Mitochondrial, Ribosomal

Funding: Institut Français du Cheval et de l’Équitation and the Fonds Éperon GeT, Toulouse, France “Investissement d’avenir” Program ANR-10-INBS-09 This work was supported by the Institut Français du Cheval et de l’Équitation and the Fonds Éperon. The GeT core facility, Toulouse, France (GeT, https://doi.org/10.15454/1.5572370921303193E12) was supported by France Génomique National infrastructure through core funding of the “Investissement d’avenir” program (contract ANR-10-INBS-09). The funders had no role in study design, data collection and analysis, decision to publish, or preparation of the manuscript.

==============================
Basic knowledge on the biology and epidemiology of equine strongylid species still needs to be improved to contribute to the design of better parasite control strategies. Nemabiome metabarcoding is a convenient tool to quantify and identify species in bulk samples that could overcome the hurdle that cyathostomin morphological identification represents. To date, this approach has relied on the internal transcribed spacer 2 (ITS-2) of the ribosomal RNA gene, with a limited investigation of its predictive performance for cyathostomin communities. Using DNA pools of single cyathostomin worms, this study aimed to provide the first elements to compare performances of the ITS-2 and a cytochrome c oxidase subunit I (COI) barcode newly developed in this study. Barcode predictive abilities were compared across various mock community compositions of two, five and 11 individuals from distinct species. The amplification bias of each barcode was estimated. Results were also compared between various types of biological samples, i.e., eggs, infective larvae or adults. Bioinformatic parameters were chosen to yield the closest representation of the cyathostomin community for each barcode, underscoring the need for communities of known composition for metabarcoding purposes. Overall, the proposed COI barcode was suboptimal relative to the ITS-2 rDNA region, because of PCR amplification biases, reduced sensitivity and higher divergence from the expected community composition. Metabarcoding yielded consistent community composition across the three sample types. However, imperfect correlations were found between relative abundances from infective larvae and other life-stages for Cylicostephanus species using the ITS-2 barcode. While the results remain limited by the considered biological material, they suggest that additional improvements are needed for both the ITS-2 and COI barcodes.

Introduction

Equine strongylids encompass a diverse fauna of 14 Strongylinae and 50 Cyathostominae described species (Lichtenfels, Kharchenko & Dvojnos, 2008). Among these, Strongylus vulgaris is responsible for the death of animals because of verminous arteritis liver pathology and peritonitis while Cyathostominae impinge on their host body condition (McCraw & Slocombe, 1976, 1978, 1985; Reinemeyer & Nielsen, 2009). In addition, the mass emergence of developing cyathostomin stages can lead to a fatal syndrome of cyathostominosis characterised by abdominal pain, diarrhoea or fever (Giles, Urquhart & Longstaffe, 1985). The release of modern anthelmintics has drastically reduced the prevalence of Strongylus spp in the field as first mentioned in 1990 (Herd, 1990) and later confirmed by observations from necropsy data (Lyons et al., 2000; Sallé et al., 2020). However, treatment failure against various cyathostomin species, primarily Cylicocyclus spp (van Doorn et al., 2014; Kooyman et al., 2016) has been found on many occasions across every continent for all drug classes currently available (Peregrine et al., 2014). Despite their worldwide distribution and relevance for stakeholders in the field, little knowledge has been gathered on the mechanisms driving their assemblage. Recent meta-analyses found that the community structure of adult strongylids was little affected by geo-climatic factors (Bellaw & Nielsen, 2020), and observations have been gathered on the relationship between larval biology and temperature both in the field or under laboratory conditions (Ogbourne, 1972; Kuzmina, Kuzmin & Kharchenko, 2006). Strongylid population structure also varies according to horse age (Torbert et al., 1986; Bucknell, Gasser & Beveridge, 1995; Kuzmina, Dzeverin & Kharchenko, 2016) or the host sex (Kornaś et al., 2010; Sallé, Kornaś & Basiaga, 2018). The tedious and delicate process of species identification by morphological keys (Lichtenfels, Kharchenko & Dvojnos, 2008) is a major hurdle to study further the mechanisms of species assemblage, their turnover and the respective impacts of the host and their environment.

DNA-metabarcoding is a non-invasive, time- and cost-effective method for assessing nematode populations that provides data with comparable taxonomic resolution to morphological methods (Avramenko et al., 2015; Redman et al., 2019; Poissant et al., 2021). This requires appropriate barcodes able to distinguish between the various phylogenetic strata. The internal transcribed spacer 2 region (ITS-2) of the nuclear rRNA gene (Blouin, 2002; Kiontke et al., 2011) and the mitochondrial COI gene (Blouin, 2002; Blaxter et al., 2005; Prosser et al., 2013) have already been used for nematode molecular barcoding. For cyathostomin species, early barcoding attempts relied on the polymorphisms present in the ITS-2 rDNA region (Hung et al., 1999, 2000) before additional contributions were made using the COI gene (McDonnell et al., 2000), or the longer intergenic spacer sequence (Cwiklinski et al., 2012). Additional work recently highlighted how the COI region could increase the resolution of species genetic diversity, suggesting a close phylogenetic relationship between Coronocyclus coronatus and Cylicostephanus calicatus (Bredtmann et al., 2019; Louro et al., 2021). In addition to this higher resolutive power, the protein-coding nature of the COI barcode can be leveraged to denoise sequencing data (Ramirez-Gonzalez et al., 2013). To date, metabarcoding experiments on strongylid species of equids have exclusively focused on the ITS-2 rDNA gene region, including community description in wild and domesticated equine populations (Poissant et al., 2021; Sargison et al., 2022), the study of bioactive forage effect on cyathostomin species (Malsa et al., 2022), the evaluation of drug efficacy in cyathostomin population (Nielsen et al., 2022) or investigation of parasitic nematode community in plain zebras (Tombak et al., 2021). This may owe to the existence of universal primers and the amplicon length that is a good fit for short-read sequencing platforms. Observations in helminths also suggest that amplification efficiency is suboptimal for the COI region (Prosser et al., 2013) which speaks against its application for metabarcoding purposes. However, mitochondrial markers have better discriminating abilities between closely related or cryptic species (Bredtmann et al., 2019; Gao et al., 2020; Louro et al., 2021), supporting the added value of the COI barcode for the metabarcoding of cyathostomin species.

In addition, metabarcoding approaches are biased in predicting relative taxon abundances (McLaren, Willis & Callahan, 2019). These biases are inherent to various biological and technical factors including the DNA treatment procedures, the different number of cells represented by each taxon (that is tightly linked to the life-stage considered for cyathostomin species), PCR specifications (cycle number) and the amplification efficiency across taxa, and the genetic diversity (including structural variants and copy numbers) of the considered barcodes within taxa (Pollock et al., 2018). Subsequent bioinformatic processing of the sequencing data, e.g., taxonomy assignment, can also induce additional biases in the community diversity prediction (Hleap et al., 2021). Because of the complexity of cyathostomin communities (Lichtenfels, Kharchenko & Dvojnos, 2008) and their close phylogenetic relationship (Hung et al., 2000; Louro et al., 2021), it is unclear how the metabarcoding approach and taxonomy assignment could provide a fair representation of their diversity. To date, the precision and recall of the metabarcoding approach applied to cyathostomins are unresolved. However, validation of this approach using the ITS-2 rDNA barcode has been performed for cattle (Avramenko et al., 2015) and ovine strongylid species (Redman et al., 2019). Although previous work showed the number of Strongylus edentatus and cyathostomin larvae was slightly underestimated by the metabarcoding approach (Poissant et al., 2021), it is yet unknown whether this approach provides a fair description of the actual cyathostomin species presence or absence, or their accurate relative abundances in their host. In this respect, observations on small ruminant trichostrongylid species found good agreement between metabarcoding applied to eggs, first or third stage larvae (Redman et al., 2019), although complimentary observations suggested that relying on first stage larvae may reduce biases associated with differential larval development rates until the third stage across species (Borkowski et al., 2020). The impact of the considered cyathostomin life-stage, i.e., eggs, larvae or adult cyathostomins, on the predictive abilities of the metabarcoding remains unknown for cyathostomin species.

In light of the current literature, it is yet to be determined whether the COI barcode could be a valuable barcode to be used for the metabarcoding of cyathostomin populations, and the correspondence between morphological observations and metabarcoding data has not been characterised in cyathostomins. Finally, variation in the results obtained from different life-stages remains unknown for cyathostomins. To address these three questions, we developed degenerate primers to amplify the COI region following a strategy successful under other settings (Elbrecht & Leese, 2017a; Elbrecht et al., 2019). We built pools of cyathostomin DNA with known composition and applied a nemabiome metabarcoding approach targeting the ITS-2 rDNA and COI gene regions. Using this design, we tested how the predictive value was affected by the number of species in the DNA mixture and compared the performances of both barcodes. We applied the metabarcoding approach to different types of biological samples to test the hypothesis that differences in species fecundity, egg hatching rate, and larval development rate could bias the community diversity.

Materials and Methods

Mock community design and DNA extraction

To compare both barcodes and to quantify biases in cyathostomin community prediction, DNA mixtures were made from single worms of 11 species at most. These mixtures are refered to as ‘mock communities’ as their compositions were known, although they do not represent typical worm communities as encountered in the field, where multiple individuals from each species would be available. This choice was grounded by (i) the limited availability of the worm material, (ii) the inability to distinguish between the role of differential worm contribution to the DNA pool and the contribution of within-species diversity to any biases in community diversity estimate after pooling worms together.

Mock communities were built from morphologically identified equine cyathostomin specimens from pooled faecal samples in Ukraine (Kuzmina, Dzeverin & Kharchenko, 2016) and Poland. For each species and community size, a single adult male was digested using proteinase K (Qiagen, Hilden, Germany) in lysis buffer, before DNA extraction using a phenol/chloroform protocol. DNA was precipitated overnight in ethanol and sodium acetate (5M) at −20 °C and washed twice with 70% ethanol. The resulting DNA pellet was resuspended in 30 µL of TE buffer (10 mM Tris-HCl, 0.1 mM EDTA, pH 8.0), and DNA was quantified using using the Qubit® double-stranded high-sensitivity assay kit (Life Technologies™, Carlsbad, CA, USA) with a minimum sensitivity of 0.1 ng/µL. Extracted DNA was stored at −20 °C.

To quantify the impact of the number of species in the community, mock communities of 11 and five species were built and subjected to amplicon sequencing using the ITS-2 rDNA and mitochondrial COI barcodes (Table 1). Within each community, species DNA was either added on an equimolar basis or at their respective concentrations to mimic differences occurring when mixing species with differential contributions (heterogeneous communities; Table 1). Of note, two Cyathostomum pateratum individuals were added in the five- and 11-species communities to assess the impact of inter-individual variation. To further measure the resolution ability of the nemabiome approach, two-species communities were made with Cyathostomum catinatum and C. pateratum. Both species were either in imbalanced ratios (three-fold difference in both directions) or equal DNA concentration.

Table 1 Detailed mock community composition.

The detailed composition of the eight mock communities used in this study is provided with the respective final DNA concentration and relative abundance of each species. Homogeneous refers to equimolar contribution of each species to the DNA pool. Heterogeneous corresponds to equal DNA volume added per species. For each species and community size, DNA was extracted from a single worm.

Mock community composition	Species (input DNA concentration; Fraction of total DNA amount)	
Homogeneous, 11 species	Cyathostomum pateratum (0.27 ng/µL; 16.7%)

Others (0.135 ng/µL; 8.3%):

Cylicocyclus ashworthi, Cylicocyclus insigne, Cylicocyclus leptostomum, Cylicocyclus nassatus;

Coronocyclus labratus, Coronocyclus labiatus;

Cylicostephanus calicatus, Cylicostephanus goldi, Cylicostephanus longibursatus;

Cyathostomum catinatum

	
Heterogeneous, 11 species	C. ashworthi (0.553 ng/µL; 4.6%), C. insigne (1.06 ng/µL; 8.8%), C. leptostomum (0.549 ng/µL; 4.54%), C. nassatus (1.28 ng/µL; 10.7%)

C. labratus (0.251 ng/µL; 2.1%), C. labiatus (0.498 ng/µL; 4.1%)

C. calicatus (0.216 ng/µL; 1.8%), C. goldi (0.135 ng/µL; 1.1%), C. longibursatus (1.34 ng/µL; 11.1%)

C. pateratum (5.14 ng/µL; 42.8%), C. catinatum (1.08 ng/µL; 8.9%)

	
Homogeneous, five species	C. pateratum (4 ng/µL; 33.3%);

Others (2 ng/µL; 16.7%): C. insigne, C. nassatus, C. labiatus, C. catinatum

	
Heterogeneous, five species	C. insigne (0.373 ng/µL; 4.1%), C. nassatus (1.07 ng/µL; 11.6%);

C. labiatus (1.52 ng/µL; 16.5%);

C. catinatum (0.64 ng/µL; 6.9%), C. pateratum (5.61 ng/µL; 60.82%)

	
Homogeneous, two species	C. catinatum (0.5 ng/µL; 25%), C. pateratum (3.5 ng/µL; 75%)

	
Homogeneous, two species, low	C. catinatum (0.135 ng/µL; 50%), C. pateratum (0.135 ng/µL; 50%)

	
One-to-four ratio, two species	C. catinatum (3 ng/µL; 75%), C. pateratum (1 ng/µL; 25%)

	
Three-to-four ratio, two species	C. catinatum (1 ng/µL; 25%), C. pateratum (3 ng/µL; 75%)

	

Parasite material collection for comparison of the metabarcoding performances across sample type

Parasite material was collected from six Welsh ponies with patent strongylid infection (Table 2). Faecal matter (200 g) was recovered from the rectum and incubated with 30% vermiculite at 25 °C and 60% humidity for 12 days before third-stage larvae were recovered using a Baerman apparatus.

Table 2 Quantity of larvae, adults and eggs used for DNA extractions.

The quantities indicated are those present in each faecal aliquot. For every test sample collected from six Welsh ponies, the quantity of recovered parasite material is indicated for each type of biological material recovered.

Host tag	Adults worms	Infective larvae	Number of eggs	
W646	50	17,500	36,000	
W710	50	24,000	43,000	
W729	50	21,000	35,000	
W733	17	20,000	35,000	
W734	50	12,000	35,000	
W748	50	27,000	35,000	

To estimate larval concentration, thirty 5-µL aliquots were sampled to count the larvae in each. Strongylid eggs were extracted from another 200 g of faeces. For this, faecal matter was placed onto a coarse sieve to remove large plant debris, before further filtering was made on finer sieves (150 µM and 20 µM mesh). Kaolin (Sigma K7375; Sigma-Aldrich, St. Louis, MO, USA) was then added (0.5% w/v) to the egg suspension to further pellet contaminating debris (5 min centrifugation at 2,000 rpm). The supernatant was discarded and the egg pellet was resuspended in a dense salt solution (NaCl, d = 1.18) and centrifuged slowly (1,200 rpm for 5 min), before this final egg suspension was placed on a 20 µM mesh sieve for the last wash. The eggs were then counted in 30 drops of 5 µL to estimate their concentration. Adult worms were collected from the same ponies at 18 and 21 h after a pyrantel embonate treatment (Strongid®; Zoetis, Malakoff, France; 6.6 mg/Kg body weight). Pyrantel efficacy was 87.5% (95% confidence interval [78.5–93.7%]) for this isolate as described elsewhere (Boisseau et al., 2022). DNA extraction was performed as for the mock community samples. All DNA concentrations were standardized to 5 µL/mL.

COI and ITS-2 primer design

We aimed to define a 450-bp amplicon within the 650-bp fragment of the cytochrome c oxidase subunit I (COI) locus previously described (Bredtmann et al., 2019; Louro et al., 2021). This would leave at most 50 bp overlap, thereby allowing sequencing error correction and better amplicon resolution (Edgar & Flyvbjerg, 2015). For this, 18 mitochondrial sequences of 17 strongylid species with complete mitochondrial genomes available at that time (October 9, 2020) were retrieved from GenBank using the PrimerMiner package v4.0.5 (Elbrecht & Leese, 2017b). They encompassed 11 species of Cyathostominae (AP017681: Cylicostephanus goldi, GQ888712: Cylicocyclus insigne; NC_032299: Cylicocyclus nassatus; NC_035003: Cyathostomum catinatum; NC_035004: Cylicostephanus minutus; NC_035005: Poteriostomum imparidentatum; NC_038070: Cyathostomum pateratum; NC_039643: Cylicocyclus radiatus; NC_042141: Cylicodontophorus bicoronatus; NC_042234: Coronocyclus labiatus; NC_043849: Cylicocyclus auriculatus; NC_046711: Cylicocyclus ashworthi) and six species of Strongylinae (AP017698 and Q888717: Strongylus vulgaris; NC_026729: Triodontophorus brevicauda; NC_026868: Strongylus equinus; NC_031516: Triodontophorus serratus; NC_031517: Triodontophorus nipponicus). These sequences were aligned with Muscle v.3.8.21 (Edgar, 2004). Subsequently, this alignment was used to quantify sequence heterozygosity for 450-bp sliding windows using a custom python script (File S1). The consensus sequence of the region with the highest diversity, i.e., best discriminant across species, was isolated to design primers with the Primer3 blast web-based interface (Untergasser et al., 2012). Parameters were chosen to have an amplicon product of 400–450 bp, primers of 20 bp with melting temperatures of 60 ± 1 °C. Primer sequences were subsequently degenerated to account for identified SNPs in the mitochondrial sequence alignment, yielding a 24-bp long forward (5′-RGCHAARCCNGGDYTRTTRYTDGG-3′) and 25-bp long reverse (5′-GYTCYAAHGAAATHGAHCTHCTHCG-3′) primers. For the ITS-2 barcode, we relied on previously described NC1 (5′-ACGTCTGGTTCAGGGTTGTT-3′) and NC2 (5′-TTAGTTTCTTTTCCTCCGCT-3′) primers applied on strongylid species (Gasser et al., 1993) and used in previous phylogenetic (Hung et al., 2000) and metabarcoding experiments (Poissant et al., 2021). In both cases, a random single, double or triple nucleotide was added to the 5′ primer end to promote sequence complexity and avoid signal saturation. A 28-bp Illumina overhang was added for the forward and reverse sequences respectively, for subsequent ligation with Illumina adapters.

Library preparation and sequencing

For library preparation, PCR reactions were carried out in 80 µL with 16 µL HF buffer 5X, 1.6 µL dNTPs (10 mM), 4 µL primer mix containing forward and reverse primers, 0.8 µL Phusion High-Fidelity DNA Polymerase (2 U/µL; Thermo Scientific, Waltham, MA, USA), and 2 µL of a 5 ng/µL genomic DNA solution quantified using a Qubit® double-stranded high sensitivity assay kit (Life Technologies™, Carlsbad, CA, USA) with a minimum sensitivity of 0.1 ng/µL. A nested PCR approach was considered to compensate for the diversity found across species over the COI region and the suboptimal amplification efficacy for the considered barcode. The COI region was first amplified using the following conditions (Duscher, Harl & Fuehrer, 2015): 95 °C for 3 min, then 30 cycles at 95 °C for 30 s, 50 °C for 30 s, 72 °C for 30 s then a final extension of 2 min at 72 °C. The resulting PCR products were then diluted at 1/70th for the second round using the degenerate primers targeting the chosen 500 bp amplicon. The PCR parameters for this second round were 95 °C for 3 min, followed by a pre-amplification with five cycles of 98 °C for 15 s, 45 °C for 30 s, 72 °C for 30 s, followed by 35 cycles of 98 °C for 15 s, 55 °C for 30 s, 72 °C for 30 s then a final extension of 2 min at 72 °C. In that case, the five-cycle pre-amplification at lower temperature were applied to compensate for suboptimal priming of the degenerate primers (Kwok et al., 1994). For comparison between the two barcodes, the number of amplification cycles was kept identical to the original application on equine strongylids (Hung et al., 2000): 95 °C for 3 min for the first denaturation, then 30 cycles at 98 °C for 15 s, 60 °C for 15 s, 72 °C for 15 s, followed by a final extension of 72 °C for 2 min.

For each sample, 20 µL were examined on 1% agarose gel to check for the presence of a PCR amplification band at the expected product size (or absence thereof for negative controls). PCR products were purified with magnetic beads (0.8X; AMPure XP, Beckman Coulter, Brea, CA, USA) following the manufacturer’s recommended protocol. A homemade six-bp index was added to the reverse primer during a second PCR with 12 cycles using forward primer (-AATGATACGGCGACCACCGAGATCTACACTCTTTCCCTACACGAC-) and reverse primer (-CAAGCAGAAGACGGCATACGAGAT-index-GTGACTGGAGTTCAGACGTGT-) for single multiplexing. The amplicons were purified using 1X Ampure XP beads (Beckmann Coulter, Brea, CA, USA) and the concentration was checked using a NanoDrop 8000 spectrophotometer (Thermo Fisher Scientific, Waltham, MA, USA). The quality of a set of amplicons was controlled using a Fragment Analyser (Agilent Technologies, Santa Clara, CA, USA).

The final libraries had a diluted concentration of 5 to 90 ng/µL, and 150 ng of each library was pooled for combined library production. Quantification was done by qPCR using the Kapa Library Quantification Kit (Roche, Basel, Switzerland) and loaded onto the Illumina V3 500 cycles MiSeq cartridge (2 × 250 output; Illumina, San Diego, CA, USA) according to the manufacturer instructions. The quality of the run was checked internally using 15% of PhiX control, and then each pair-end sequence was assigned to its sample with the help of the previously integrated index.

Analytical pipelines

Quality control and filtering

Sequencing data were first filtered using cutadapt v1.14 to remove insufficient quality data (-q 15), trim primer sequences, and remove sequences with evidence of indels (—no-indels) or that showing no trace of primer sequence (—discard-untrimmed).

Analytical pipelines for community structure inference using the ITS-2 amplicon

The implemented framework was similar to previous work (Poissant et al., 2021) that used the DADA2 algorithm (Callahan et al., 2016) to identify amplicon sequencing variants after error rate learning and correction. The denoising procedure, consisting in learning error rates independently for both forward and reverse reads, was applied for two discrete stringency parameter sets either tolerating a single error for both reads (mxEE = 1) or more relaxed stringency (mxEE = 2 and 5 for the forward and reverse reads respectively). The truncation length of forward and reverse reads was set to 200 (shorter reads retained, smaller overlap for merging but further from the 3′ end with lower quality) or 217 bp (minimal read length in our data) to promote merging between the reads. A minimal overlap of 12 bp was set and no mismatch tolerated (default value) between both reads. Last, the band_size parameter effect was also explored considering three values, i.e., −1, 16 and 32, that respectively disable banding and implement the default or a more relaxed value that is recommended for amplicon of variable length (Callahan, McMurdie & Holmes, 2021) as is the ITS-2 sequence in equine strongylid (Hung et al., 2000). In every case, denoising was run using the pseudo-pool option, and chimera detection relied on the consensus mode. The ‘pseudo-pooling’ was chosen over ‘pooling’ for denoising to establish priors for each sample independently, thereby accounting for the different library structures (varying between two and 11 species at most).

Taxonomic assignment was subsequently performed using a sequence composition approach using the IDTAXA algorithm (Murali, Bhargava & Wright, 2018) as implemented in the DECIPHER R package v.2.18.1 with minimal bootstrap support of 50%. This last step relied on the ITS-2 rDNA database for nematodes v.1.3.0 (Workentine et al., 2020). Building on previous work (Poissant et al., 2021), 22 truncated sequences and a dubious Cyathostomum catinatum entry (accession number Y08619.1) were removed from the whole database which was left untouched otherwise. In the end, 264 equine strongylid sequences were available for analysis.

Analytical pipelines for community structure inference using the COI amplicon

For the COI barcode, we built a custom COI barcode sequence database for Cyathostominae and Strongylinae species collected from Genbank, BOLD database using the Primerminer package v.0.18 (Elbrecht & Leese, 2017b) and concatenated into a single fasta file. Sequences were subsequently edited to remove elephant Cyathostominae species (Quilonia sp, Murshidia sp, Kilonia sp, and Milulima sp) using the seqtk v.1.3 subseq option (https://github.com/lh3/seqtk). Some entry names (n = 18) consisted of an accession number that was manually back-transformed to the corresponding species name. Duplicate entries were removed with the rmdup option of the seqkit software v.0.16.0 (Shen et al., 2016) and sequences were dereplicated using the usearch v11.0.667 -fastx_uniques option (Edgar, 2010). To reduce the database complexity and promote primary alignment, the most representative sequences were further determined using the usearch -cluster_smallmem option, considering two identity thresholds of 97% and 99%. The final database consisted of 241 sequences corresponding to 32 equine strongylid species (seven large strongyle species), including all species considered further in our mock communities.

Amplicon analysis relied on a mapping approach to the custom COI sequence database implemented using the minimap2 software v.2-2.11 (Li, 2018) as described elsewhere (Ji et al., 2020). First, paired-end reads were merged into amplicon sequences using the usearch software v11.0.667 and the -fastq_mergepair option (Edgar, 2010). Merged amplicon sequences were subsequently mapped onto the COI sequence database using the minimap2 (Li, 2018) short read mode (-ax sr). Mapping stringency was varied to select the most appropriate combinations using k-mer sizes of 10, 13, and 15 (default), window sizes w of 8, 9 or 10 (default), and varying mismatch penalty values (B = 1, 2, 3 or the default values 4). The lower the value, the more permissive the alignment for these three parameters. Produced alignments were converted to bam files using samtools v1–10 after filtering against unmapped reads, alignments that were not primary and supplementary secondary alignments using the -F 2308 flag (Li et al., 2009). To evaluate how mapping stringency, filtered bam files were also produced using a mapping quality cut-off of 30. Species abundance was then inferred from read depth over each COI sequence that was determined using the bedtools genomecov algorithm (Quinlan & Hall, 2010) and scaled by the sequence length.

Quantitative PCR (qPCR) assay for species-specific amplification

To quantify any biases in PCR amplification before sequencing, single-species DNAs used to make the mock communities were subjected to quantitative PCR reactions with the ITS-2 and COI-specific primers. Other worms collected following pyrantel treatment in an experimental pony herd were also used for qPCR to avoid relying on a single sample. Their DNA was used for sequencing of the ITS-2 and COI barcodes and belonged to the C. ashworthi, C. catinatum, C. goldi, C. labiatus, C. leptostomum, C. longibursatus, C. nassatus and C. pateratum species. One male and one female were available, but for C. catinatum and C. longibursatus (single male collected).

The DNA was diluted at 1:250 and 1 µL of DNA was added to each reaction. qPCRs were carried out on a Biorad CFX Connect Real-Time PCR Detection System following the iQ SYBR GREEN supermix® protocol (1708882; Biorad, Marnes-la-Coquette, France). Reactions were run in triplicate for each species with 40 amplification cycles: 95 °C for 3 min for the first denaturation, then 45 cycles of 98 °C for 15 s, 60 °C for 30 s, 72 °C for 40 s, followed by a melt curve (65 °C to 95 °C).

Statistical Analyses

Statistical analyses were run with R v.4.0.2 (R Core Team, 2016). Community compositional data were imported and handled with the phyloseq package v1-34.0 (McMurdie & Holmes, 2013). Abundance data (read count for the ITS-2 rDNA region, or scaled read depth for COI) were aggregated at the species level using the taxglom() function of the phyloseq package v1-34.0 and converted to relative abundances for further analysis. These data were used to compare each barcode and pipeline predictive ability in the first comparison. After the most appropriate bioinformatics parameters were identified for each barcode, ASVs with outlier representation, i.e., less than 100 reads per base pair for the COI barcode or 40 reads for the ITS-2 rDNA barcodes, were regarded as likely contaminants and further discarded.

Species richness, alpha-diversity and beta-diversity analyses were conducted with the vegan package v2.5-7 (Oksanen et al., 2017). PerMANOVA was implemented using the adonis() function of the vegan package v2.5-7 (Oksanen et al., 2017).

To monitor the predictive ability of each pipeline and barcode, the precision (the proportion of true positives among all positives called, i.e., true positives and false positives) and recall (the proportion of true positives among all true positives, i.e., true positives and false negatives) of the derived community species composition were computed and combined into the F1-score as:

F1=2×recall×precisionrecall+precision

This score supports the ability of a method to correctly identify species presence while minimising the number of false-positive predictions, i.e., a trade-off between precision and recall (Hleap et al., 2021).

Alpha diversity was estimated using the Shannon and Simpson’s indices using the estimate_richness() function of the phyloseq package (McMurdie & Holmes, 2013). The difference between the expected mock community expected and observed alpha diversity values was further considered to compare conditions and barcodes. The divergence between the inferred community species composition and the true mock community composition was estimated from a between-community distance matrix based on species presence/absence (Jaccard distance) or species relative abundances (Bray-Curtis dissimilarity) using the vegdist() function of the vegan package (Oksanen et al., 2017). Compositional differences between sample types were visualised with a non-metric multidimensional scaling (NMDS) with two dimensions using Jaccard and Bray-Curtis dissimilarity.

For each of these variables (F1-score, alpha-diversity differences and divergence) and within each barcode, estimated values were regressed upon bioinformatics pipeline parameters (mxee, truncation length and band size parameters for the ITS-2 barcode data; k-mer size, window size, mismatch penalty and mapping quality for the COI barcode) to estimate the relative contribution of these parameters and determine the most appropriate analytical pipeline for each barcode independently. The model with the lowest Akaike Information Criterion value was first selected with the stepAIC() function of the R MASS package v.7.3-55 (Venables & Ripley, 2002) to retain the most relevant parameter combination (model with the lowest AIC). Parameter values were then chosen according to their least-square mean estimate. These analyses were applied to every available data within each barcode.

These variables were subsequently regressed upon barcodes and the mock community complexity (two species vs five species and more) to estimate how the predictive performances were affected by these predictors. Correspondence between input DNA and recovered reads was estimated using Spearman’s correlation applied to the homogeneous and heterogeneous communities separately (Table 1).

To test for differences in alpha diversity across sample type (eggs, larvae or adult worms), the Shannon index was regressed upon the sample type and barcode using the lm() function.

To estimate amplification efficiencies, the threshold cycle (Ct) values were regressed upon the log10-transformed DNA concentration for each species and barcode. The PCR efficiency was subsequently derived as: Ei,j=10−1βi,j, where Ei,j is the efficiency, and βi,jstands for the regression slope of species i and barcode j.

The R script and the necessary files used for analysis are available under the INRAE data repository at https://doi.org/10.57745/MNYRFQ.

Results

Impact of the community size on the ITS-2 and COI barcode performances

To ensure a fair comparison, the most appropriate bioinformatic processing was determined for each barcode according to their predictive performances of mock community composition (Supplemental Information and Tables S1–S3). This comparison relied on the same two mock communities of five cyathostomin species (Table 1). The combination of the minimap default values for the mismatch penalty (B = 4) and the window size (w = 10) parameters, with a k-mer size of 10 base pairs and no further filtering on mapping quality (MQ = 0) was deemed as the most appropriate pipeline for the COI barcode in this study. For ITS-2, stringent tolerance in the maximal expected number of errors and a truncation length of 200 bp were chosen for downstream analyses.

With these settings, the number of reads available for the considered mock communities ranged between 6,164 and 151,606 read pairs for the COI barcode, while these numbers ranged between 1,243 and 88,845 non-chimeric pairs for the ITS-2 rDNA barcode.

No significant difference was found in the F1-score between the ITS-2 and COI barcodes overall (F1,19 = 0.26, P = 0.61; Fig. 1). However, higher predictability was obtained with the ITS-2 barcode for the more complex community as suggested by the significant interaction term between mock community size and barcode predictors (t1 = 2.7, P = 0.02; Fig. 1). Regarding species detection, the COI barcode ability to recover the true species composition was suboptimal. This barcode recovered nine correct species at most in the most complex communities and systematically overlooked C. leptostomum and C. labratus. On the contrary, it also identified C. coronatus (in all of the four 11-species mock communities) and C. minutus (in one out of the four 11-species mock communities) despite these species absence. Their relative abundances remained however lower than 1% for C. coronatus and 0.02% for C. minutus and were associated with low mapping quality (Phred quality score <6). The ITS-2 rDNA barcode performed better with ten species detected overall, but C. goldi was systematically overlooked in the 11-species mock communities.

Figure 1 Comparison of the predictive abilities of cyathostomin community structure for the mitochondrial COI and ITS-2 rDNA barcodes.

Considered coefficient values are represented across three mock community sizes for the mitochondrial COI (blue) and ITS-2 rDNA (yellow) barcodes. F1-score corresponds to the trade-off between identifying true positives while minimizing the false discovery rate (A). Divergence was computed as the Bray-Curtis (species relative abundances); (B) between the expected and observed mock community composition. Differences between observed and expected alpha diversity (Shannon’s index) are given in (C). (D) Depicts the fraction of reads with no taxonomy assigned (note that because of the stringency of the mapping procedure for the COI barcode, the rate of taxonomy assignment is inflated).

The ITS-2 barcode gave better representations of the most complex cyathostomin community composition (with five or 11 species; Fig. 1B). This superiority was observed for species relative abundances (reduction of 0.34 ± 0.14 in Bray-Curtis dissimilarity relative to COI, P = 0.03). Still, it was milder for species presence/absence (decrease of 0.32 ± 0.16 in Jaccard distance relative to COI, P = 0.07). The ITS-2 barcode also gave the closest estimates of the expected alpha diversity (F1,12 = 16.6, P = 1.5 × 10−4 and F1,12 = 15.9, P = 1.7 × 10−4 for Simpson and Shannon indices; Fig. 1C). These differences vanished, however, when considering the mock communities composed of two Cyathostomum species (Fig. 1, P > 0.2 for the six parameters considered).

Last, the fraction of unassigned reads decreased with the mock community size (Spearman’s correlation coefficient ρ = −0.64, P = 0.02, n = 13) for the ITS-2 rDNA barcode (from 18.6% to 2.51% for the two- and 11-species communities, Fig. 1D and Table S2). Because of the mapping procedure applying to the COI region, this fraction remained negligible (2 × 10−5% for the most complex community and null otherwise; Fig. 1D). In summary, none of the two barcodes offered a perfect fit for the expected community composition, but the ITS-2 rDNA barcode was more accurate for the compositional description of the communities.

PCR amplification bias for the COI and ITS-2 rDNA barcodes

The imperfect match between the mock community and the prediction from the metabarcoding approach might relate to biases in the first PCR amplification. To test this hypothesis, qPCRs were performed on each species DNA from the same single individual used for library preparation (Fig. 2, and Table S3). The average amplification efficiency was 67.7 ± 0.24% for the COI barcode. It was above 90% for the two Cyathostomum species (Table S3) but it fell below 70% for five species. Among these, C. calicatus and C. longibursatus showed the lowest values (39% and 13% respectively, Fig. 2 and Table S3).

Figure 2 Species-specific amplification efficiencies of the COI and ITS-2 barcodes for 11 cyathostomin species.

The amplification efficiency (in %) derived from qPCR is plotted for each species of interest. Each dot represents a single worm. Worms from the first experiment corresponds to the same batch as those used for metabarcoding sequencing, while the second experiment corresponds to a validation set. Dot shape indicates the worm sex.

Omitting the outlier values found for the Cylicostephanus members (C. goldi undetected and too high efficiency for C. calicatus; Fig. 2 and Table S3), the ITS-2 rDNA barcode yielded more consistent and higher amplification efficiency on average (92.2 ± 0.03%; t10 = 3.42, P = 0.006) than COI.

To confirm that the results were not specific to the considered set of worms, independent qPCR were run for male and female worms (Fig. 2 and Supplemental 3). In that case, the ITS-2 rDNA barcode showed efficiencies above 93% for every species and sex. However, the efficiency for the COI barcode dropped to 38 ± 28% on average (Fig. 2 and Supplemental 3).

The considered life stage has limited effect on the inference of community diversity

Differential worm fecundity or larval development may also contribute to distort the correlation between relative abundances measured from different sample types for some species. To test this hypothesis, nemabiome metabarcoding data were generated from six horses for three life stages of strongylids (eggs, infective larvae and adult worms collected after pyrantel treatment). The total number of species present in this population remains unknown to date. The COI barcode retrieved 12 species, of which C. radiatus was not found with the ITS-2 barcode. This latter barcode allowed the retrieval of 13 species and four additional amplicon sequence variants that were assigned at the genus level only (Fig. 3). ITS-2 data also recovered C. leptostomum and Craterostomum acuticaudatum (Strongylinae) that were not found with the COI marker (Fig. 3). The fraction of unassigned reads was 2.01% on average (range between 0% and 4.28%) and did not differ across sample types (1.6% for larval samples, 1.95% for egg samples and 2.46% for adult samples).

Figure 3 Impact of the considered life-stage on predicted equine strongylid community diversity.

(A) The relative abundances measured in the strongylid communities from six horses using the COI or ITS-2 barcodes applied to either eggs, infective larvae or adult worms. (B and C) The first two axes of a non-linear multidimensional scaling analysis based on Bray-Curtis dissimilarity for the COI and ITS-2 rDNA region respectively. Unclassified sequences were not visualized for the ITS-2 barcode but accounted for 2.01% of read counts on average (range between 0% and 4.28%). Because of the outlying community structure found for ITS-2 on the larval samples of the W734 and W748 ponies, the NMDS was applied on only four individuals.

The community structure was generally in good agreement across sample types for both the ITS-2 and COI barcodes, although two larval samples exhibited outlying behaviours with the ITS-2 barcode (Fig. 3A). As a result, the samples from these two ponies were not considered further for analyses of the diversity with the two barcodes. The Shannon index showed no significant variation across the considered life stages for both barcodes (P = 0.91, F2,20 = 0.1) as the observed differences fell below the resolutive power of this experiment (difference of 1.8 detectable with a significance level of 5% and power of 80%). Shannon index estimates obtained with the ITS-2 barcode were higher (difference of 0.50 ± 0.19 relative to COI, Student’s t1 = 2.56, P = 0.019).

In agreement with this observation, PerMANOVA showed that the sample type was not a significant driver of the beta-diversity, explaining 10.6% (P = 0.53, F2,11 = 0.53) and 6.42% (P = 0.98, F2,11 = 0.31) of the variance in species relative abundances for the COI and ITS-2 barcode respectively (Figs. 3B and 3C). The same applied when considering the Jaccard distance on species presence/absence, whereby the sample type explained 12.3% (P = 0.87, F2,11 = 0.63) and 8.92% (P = 0.99, F2,11 = 0.44) of the variance for the COI and ITS-2 barcode respectively (Fig. S1).

The consistency of species relative abundance between sample types varies across cyathostomin genera

Differential larval development and fecundity may affect the correlations between different sample types. Relying on the ITS-2 based data, i.e., the most reliable barcode in our setting, correlations between species relative abundances from the three samples types were estimated for each genus. The highest consistency between inferred relative abundances was found between the egg and the larval samples (Pearson’s r = 0.96, P < 10−4, n = 64).

On the contrary, the values obtained using adult samples were less well correlated to the two others (Pearson’s r = 0.77 and 0.75 between adult worms and eggs or infective larvae respectively, P < 10−4, n = 64). These correlations were high for Cylicocyclus spp (Pearson’s r between 0.97 and 0.99, n = 20), intermediate for Cyathostomum spp and poor for both Coronocyclus spp and Cylicostephanus spp (Table 3). In the lack of major PCR amplification biases found, this may owe to differential larval development for members of the Coronocyclus spp and Cylicostephanus spp. In both cases, significant correlation was found between the eggs and larvae (Pearson’s r = 0.98 and 0.89 for Coronocyclus spp and Cylicostephanus spp respectively, Table 3) but not between the adult worms and the other two stages (P > 0.14 in all cases, Table 3). While this may result from a lack of power for Coronocyclus spp (n = 8), the number of observations was similar between Cylicocyclus spp and Cylicostephanus spp.

Table 3 Overall and genus-wise correlations between the relative abundances estimated from three sample types using the ITS-2 barcode.

The Pearson’s correlation coefficients estimated between the relative abundances inferred from the eggs, the larvae or the adult worms using the ITS-2 barcode. The correlations are either shown across the overall experiment or by genus, with the number of observations (four horses times the number of species) available presented in each case.

		Eggs	L3	Adults	
Overall	Eggs	1	0.96	0.75	
n = 96	L3		1	0.77	
	Adults			1	
Coronocyclus spp	Eggs	1	0.98	0.15	
n = 8	L3		1	0.17	
	Adults			1	
Cyathostomum spp	Eggs	1	0.99	0.60	
n = 12	L3		1	0.64	
	Adults			1	
Cylicocyclus spp	Eggs	1	0.99	0.97	
n = 20	L3		1	0.97	
	Adults			1	
Cylicostephanus spp	Eggs	1	0.89	0.28	
n = 20	L3		1	0.42	
	Adults			1	
Note:

Statistically significant correlations are highlighted in bold and italicised.

Discussion

This study attempted to develop a new barcode based on the COI gene and compared its predictive performances to the rDNA ITS-2 sequence. Overall, the proposed COI barcode appears suboptimal in comparison to the ITS-2. A community of known cyathostomin composition was built to determine the most appropriate bioinformatic pipeline parameters. Comparing the results between different life-stages suggests that ITS-2 and COI metabarcoding are robust across the sample types considered. However, the ITS-2 based relative abundances of Cylicostephanus spp and Coronocyclus spp estimated using adult worms depart significantly from that found with eggs and larvae.

To compare both barcodes and their predictive performances, DNA mixtures were made of single worms due to the scarcity of available material. We considered both equimolar pools or worms as the unit of composition as applied previously (Avramenko et al., 2015). While this strategy provides a common ground to evaluate the predictive performances of both barcodes and bioinformatic processing, the results presented may be confounded by the unique properties of the individuals chosen for the experiment. Additional experiments with DNA mixtures made from multiple individuals per cyathostomin species would be needed to quantify further the putative biases associated with this diversity.

The chosen barcodes differed in many aspects. The COI region has been used for nemabiome or phylogenetic studies of other nematode species, including Haemonchus contortus (Blouin, 2002) or some free-living Caenorhabditis species (Kiontke et al., 2011). Its higher genetic variation poses this marker as an ideal barcode for cyathostomin species with evidence that it could better delineate the phylogenetic relationship between Coronocyclus coronatus and Cylicostephanus calicatus members (Louro et al., 2021). This study aimed to produce amplicons suitable for merging of read pairs, i.e., with a total length lower than 500 bp. The chosen strategy (including degenerate primers, pre-amplification step and lower mapping stringency) yielded a poor correlation between input DNA and quantified reads. This certainly owes to amplification biases associated with the multiple amplification steps and a reduced specificity of the lower mapping stringency. However, past experiments dealing with arthropods (Krehenwinkel et al., 2017) or microbial species (Sipos et al., 2007) did not find evidence of significant biases due to the increased number of PCR cycles. On the contrary, the universal ITS-2 primers yield an amplicon suitable for paired-end sequencing with a substantial overlap between both fragments. This combined with consistent amplification efficiency across species makes it more suitable than the chosen COI barcode for metabarcoding purpose. Besides, the development of an extensive ITS-2 database offers an easy-to-use tool for taxonomy assignment that covers a wide breadth of the equine strongylid diversity. For the COI barcode, the gathered sequences encompassed fewer species than the ITS-2 database, although all the species of interest were included.

The PCR amplification showed slight variation for the ITS-2 but these were not validated in the independent samples used for validation. Of note, additional factors such as the type of DNA polymerase could also affect the metabarcoding approach and warrant further investigations (Nichols et al., 2018).

The designed COI barcode did not outperform the ITS-2 rDNA region in terms of species detection or Jaccard-based (species presence or absence) diversity estimates. Nonetheless, inclusion of C. coronatus in mock communities may have yielded less favourable results for the ITS-2 rDNA. Indeed, this species entertains a close phylogenetic relationship with C. calicatus that may decrease the rate of correct taxonomy assignment (Bredtmann et al., 2019). In this respect, the COI barcode would offer additional specificity and should be considered further for the metabarcoding of cyathostomin species through other strategies. It remains unclear however why C. leptostomum and C. labratus showed good amplification in the first batch but were not found by the metabarcoding approach. This may indicate mis-assignation of sequencing reads to other closely related species.

While the described approach was suboptimal, other strategies targeting the mitochondrial genome could still be applied (Liu et al., 2016; Ji et al., 2020). First, bulk shotgun sequencing of cyathostomin populations could be used for mapping against a reference database of mitogenomes as applied to arthropods (Ji et al., 2020). This strategy improved the correlation between species input DNA and the number of mapped reads (Ji et al., 2020). This is however more expensive and is limited by the available mitogenomic resources for equine strongylids (17 species available at the time of this experiment). Primer cocktail to simultaneously amplify multiple amplicons is another alternative that may increase the range of diversity being sampled (Chase & Fay, 2009). However, its performance under other settings was poor and was not better than relying on degenerate primers (Elbrecht et al., 2019). Last, third-generation sequencing technologies like the Pacific Biosciences and Oxford Nanopore Technologies which are both able to sequence long DNA fragments, could recover the whole COI gene or the mitochondrial genome from a pool of strongylid species. The portable Oxford Nanopore Technologies device offers a convenient set-up that can be deployed in the field (Quick et al., 2015), and could deliver full-length COI barcode data for up to 500 insect specimens in a single run (Srivathsan et al., 2018). This comes, however, at the cost of sequencing errors associated with insertion/deletion errors over homopolymeric regions (Srivathsan et al., 2018). But this drawback can be overcome as the protein-coding nature of the COI gene provides a solid basis for error denoising (Ramirez-Gonzalez et al., 2013; Andújar et al., 2018).

The predictive ability of the ITS-2 rDNA region has already been validated using mock communities of ruminant trichostrongylid species (Avramenko et al., 2015; Redman et al., 2019). For equine strongylid communities, a recent contribution reported repeatability ranging from 47% to 48% for this approach, and it established the first indications of the predictive ability for the ITS-2 rDNA region by comparing Strongylus spp abundances inferred from molecular and morphological data (Poissant et al., 2021). Here, we aimed to expand this work to characterise the metabarcoding predictive performances against mock equine cyathostomin communities. Its amplification was more robust than already reported (Louro et al., 2021) and the PCR amplification efficiency was consistent across the considered species.

Despite the breadth of the species considered in this work, it was not possible to cover every known member of the equine Cyathostominae and Strongylinae subfamilies. Past investigation has focused on Strongylus spp showing that the metabarcoding was overestimating the true abundances of S. vulgaris and S. edentatus relative to the morphological observations while the opposite was true for S. equinus and other cyathostominae (Poissant et al., 2021). Among the Cyathostominae subfamily, species of intermediate to low abundance and prevalence like C. coronatus and C. radiatus (Ogbourne, 1976; Bucknell, Gasser & Beveridge, 1995; Kuzmina, Dzeverin & Kharchenko, 2016) should be considered for further studies. Specifically, the ability of the various approaches and algorithms to delineate between C. calicatus and C. coronatus members should be further investigated. In addition, the relative abundance of C. goldi was low and the qPCR experiment suggested that worm material may have caused a suboptimal PCR amplification. However, this species is among the most prevalent cyathostomin species in the strongylid community and its estimated mean relative abundance is 6% (Bellaw & Nielsen, 2020). This estimate matched observations made in Kentucky horses using the nemabiome metabarcoding approach (Poissant et al., 2021). In contrast, independent reports using the nemabiome metabarcoding approach found C. goldi as a minor species, either absent (Sargison et al., 2022) or with relative abundance of 0.2% (Malsa et al., 2022), 1.88% and 0.03% in Canadian horse populations (Poissant et al., 2021). We suppose that the low fecundity of most Cylicostephanus species including C. goldi (Kuzmina et al., 2012) can explain why the species that actually dominated in the horse strongylid community are underestimated in nemabiome metabarcoding studies when DNA is extracted from pools of eggs or larvae. The low correlations estimated between the three sample types for Cylicostephanus spp would support this hypothesis. In addition, the comparison of morphological and molecular data applied in a drug efficacy trial suggests discrepancies between the two approaches (Nielsen et al., 2022). In the lack of consistent amplification biases presented in this study, this variation may be compatible with errors in taxonomy assignment for C. goldi. The difference in relative abundances reported elsewhere (Nielsen et al., 2022) for other species like C. longibursatus (ranked 3rd and 11th for the morphological and molecular approaches), also warrants additional investigation for this genus.

As our observations suggest that pipeline performances were dependent on the species number, an investigation of more complex mock communities, that would better reflect field samples with multiple worms per species, also remains to be completed.

In turn, the nemabiome metabarcoding is expected to unravel yet unknown facets of cyathostomin phenology, like their seasonal preference, or any priority effects between species. The overall lack of significant differences between the considered sample types (cyathostomin eggs extracted from the faecal matter, infective larvae harvested after egg culture or adult worm collection after treatment) for this approach supports a flexible implementation in the field.

Infective larvae certainly can be harvested between 10 to 14 days after sample collection and as such, they remain the most convenient sample type for field work. In contrast, egg samples will develop into first-stage larvae within 24–48 h and adult collection will be dependent on the species drug sensitivity. Using a reverse line blot assay, Cylicocyclus members were shown to re-appear more quickly after ivermectin and moxidectin treatment (van Doorn et al., 2014). This observation was confirmed in a study employing both morphological and molecular identification with indication that two Cylicocyclus species, namely C. insigne and C. nassatus were less sensitive to ivermectin and moxidectin respectively (Nielsen et al., 2022). In addition, differential sensitivity to macrocyclic lactones was observed between C. minutus, C. pateratum, and C. longibursatus (Nielsen et al., 2022). In this study, we relied on pyrantel treatment whose efficacy is still high in the population of interest (Boisseau et al., 2022) and worms were collected 24 h after treatment as described in past studies (Kuzmina et al., 2005; Sallé, Kornaś & Basiaga, 2018). As such, the recovered adult specimens are expected to give a fair representation of the worm population. Nonetheless, biases may still occur in the species relative contributions to the DNA pool because of noticeable differences in nematode sizes (Lichtenfels, Kharchenko & Dvojnos, 2008). Of note, some of the collected larval samples yielded aberrant community compositions in our study. In the lack of clear technical biases associated with the two outlier samples, i.e., similar sequencing output or DNA concentrations, it remains unclear whether this discrepancy reflects a true biological feature, e.g., biased larval development, or other technical biases like the presence of PCR inhibitor.

To overcome the described challenges owing to barcode properties, metagenomic shotgun sequencing on DNA extracted from faeces could resolve the horse gut biodiversity in a single-pot experiment. While gut microbial gene catalogues have been built recently (Mach et al., 2022; Ang et al., 2022), the genomic resources for equine strongylids remain restricted to a few mitochondrial genomes that span the Coronocyclus (Yang et al., 2020), Cyathostomum (Wang et al., 2020), Cylicocyclus (Gao et al., 2017b), Cylicostephanus (Gao et al., 2020), and Triodontophorus (Gao et al., 2017a) genera, and a single heavily fragmented genome assembly for C. goldi (International Helminth Genomes, 2019).

Conclusion

This work compared the predictive ability of the ITS-2 region and the mitochondrial COI barcode for the study of cyathostomin communities. Overall, the COI barcode developed herein was suboptimal relative to the ITS-2 region with lower recall and precision, higher divergence from the true community structure. The amplification efficiency was higher and more consistent for ITS-2. Cyathostomin larvae appear to be the most accessible biological material for metabarcoding. However, reliance on eggs extracted from the faecal matter or adult worms yielded similar results and could be considered for studies. Overall, metabarcoding with the ITS-2 rDNA barcode gives a fair representation of the communities although Cylicostephanus species might be under-represented.

The use of a DNA pool of known species composition can support the choice of appropriate bioinformatic parameters for the study of cyathostomin communities. Additional investigation is needed to characterized the effect of cyathostomin diversity on the ITS-2 based metabarcoding approach and other strategies are needed to make use of the COI barcode for cyathostomins.

Supplemental Information

Supplemental Information 1 Sensitivity of the COI barcode across a range of various bioinformatic parameters.

For every combination of bioinformatic parameters, the number of species, true positive detection, false positive detection and the B, k and w parameters values of the minimap software are listed.

Click here for additional data file.

Supplemental Information 2 Predictive ability of the COI barcode according to chosen bioinformatic parameters.

For every bank and parameter combination, the richness, true positive, false positive, true negative, false negative, recall, precision, F1-score, true positive rate, flase positive rate, true negative rate, false negative rate, barcode, number of species in the mock community, Bray-Curtis divergence, Jaccard divergence, difference from expected Simpson’s alpha diversity, difference from expected Shannon alpha diversity, the fraction of unassigned reads are listed.

Click here for additional data file.

Supplemental Information 3 Species-specific amplification efficiencies of the COI and ITS-2 barcodes for 11 cyathostomin species.

Regression coefficients (slope, associated standard error and intercept) of the amplification efficiency (measured with Ct) upon DNA concentration are given for every cyathostomin species considered. Derived amplification efficiency and regression explanatory power are also provided. Data are ordered by genus and efficiency.

Click here for additional data file.

Supplemental Information 4 Accession numbers of raw sequence data used in this work.

Click here for additional data file.

Supplemental Information 5 How bioinformatic parameters were chosen for downstream analysis as shown under main article text.

Click here for additional data file.

Supplemental Information 6 Python script for producing sequence heterozygosity of the COI region.

Click here for additional data file.

Supplemental Information 7 The first two axes of a Non-linear Multidimensional Scaling analysis based on species presence or absence (Jaccard’s distance) for the (A) COI and (B) ITS-2 rDNA region respectively.

Click here for additional data file.

The authors are grateful to the INRAE UEPAO equine facility for their support in implementing this experiment. We are grateful to the genotoul bioinformatics platform Toulouse Occitanie (Bioinfo Genotoul, https://doi.org/10.15454/1.5572369328961167E12) for providing computing and storage resources.

Additional Information and Declarations

Competing Interests

Author Contributions

DNA Deposition

Data Availability

The authors declare that they have no competing interests.

Élise Courtot performed the experiments, prepared figures and/or tables, authored or reviewed drafts of the article, and approved the final draft.

Michel Boisseau performed the experiments, prepared figures and/or tables, authored or reviewed drafts of the article, and approved the final draft.

Sophie Dhorne-Pollet performed the experiments, authored or reviewed drafts of the article, and approved the final draft.

Delphine Serreau performed the experiments, authored or reviewed drafts of the article, and approved the final draft.

Amandine Gesbert performed the experiments, authored or reviewed drafts of the article, and approved the final draft.

Fabrice Reigner performed the experiments, authored or reviewed drafts of the article, and approved the final draft.

Marta Basiaga performed the experiments, authored or reviewed drafts of the article, and approved the final draft.

Tetiana Kuzmina performed the experiments, authored or reviewed drafts of the article, and approved the final draft.

Jérôme Lluch performed the experiments, authored or reviewed drafts of the article, and approved the final draft.

Gwenolah Annonay performed the experiments, authored or reviewed drafts of the article, and approved the final draft.

Claire Kuchly performed the experiments, authored or reviewed drafts of the article, and approved the final draft.

Irina Diekmann performed the experiments, authored or reviewed drafts of the article, and approved the final draft.

Jürgen Krücken conceived and designed the experiments, authored or reviewed drafts of the article, and approved the final draft.

Georg von Samson-Himmelstjerna conceived and designed the experiments, authored or reviewed drafts of the article, and approved the final draft.

Nuria Mach conceived and designed the experiments, authored or reviewed drafts of the article, and approved the final draft.

Guillaume Sallé conceived and designed the experiments, performed the experiments, analyzed the data, prepared figures and/or tables, authored or reviewed drafts of the article, and approved the final draft.

The following information was supplied regarding the deposition of DNA sequences:

The raw sequence data are available at SRA: PRJNA840924, SRR19334727 to SRR19334699 and PRJNA849212 (Table S4).

The following information was supplied regarding data availability:

The scripts and associated preprocessed data and outputs are available at the INRAE data repository: Sallé, Guillaume, 2023, “Horse nemabiome benchmark”, https://doi.org/10.57745/MNYRFQ, Recherche Data Gouv, V1.

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
