# Peer review of "Comparison of two molecular barcodes for the study of equine strongylid communities with amplicon sequencing"

_PeerJ, doi:10.7717/peerj.15124_

## Round 0.1 · original submission · Major Revisions

Dear Dr. Courtot and colleagues:

Thanks for submitting your manuscript to PeerJ. I have now received two independent reviews of your work, and as you will see, the reviewers raised some concerns about the research. Despite this, these reviewers are optimistic about your work and the potential impact it will have on research studying equine microbiomes and nematode parasites. Thus, I encourage you to revise your manuscript, accordingly, taking into account all of the concerns raised by both reviewers.

While both reviewers find your work relevant, particularly the use of a new barcoding tool, there is concern over several aspects of your manuscript. Please provide references to the use of imperfect tests (discussion). While you rightly expose the knowledge gap, your own attempts to apply an additional marker (COI) highlight the challenge in the field. Also, the idea of complete genomes for strongylids is many years away and important questions in cyathostomin biology are being asked in the present. Please make it clear the scope and interpretations of your study are heavily limited.

Please note that Reviewer 2 kindly provided a marked-up version of your manuscript.

I look forward to seeing your revision, and thanks again for submitting your work to PeerJ.

Best,

-joe

Reviewer 1 ·

Basic reporting

I commend the authors for a very well conducted study on the importance of using mock cyathostomin communities to clarify the use of ITS-2 to detect species composition in complex infections. The inclusion of a second barcode, the CO1, was welcomed and thorough evaluation of its utility (or lack of) will be of considerable value within the field. The approach taken appears robust and well described with all relevant analyses and datasets well defined. It was a very well written paper that was enjoyable to read.

If there is one minor weakness of the proposal it is perhaps a lack of reference to the importance of working with tools that lack the precision to detect common species such as Cylicostephanus goldi as a means to promote better and more comprehensive tools. I realise the remit of PeerJ does not encompass the need for 'impact' to be clearly articulated, however i think the authors need only include a sentence or two in addition to what they already have. The introduction did a good job of setting the study in context and highlighted how these techniques are crucial to understanding the biology and epidemiology of cyathostomins. In the discussion , there could perhaps be a sentence or two added about the importance of current methodology (ITS-2 nemabiome) 'missing' certain species. I appreciate the direction of travel and conclusion of the paper is to develop more accurate tools using metagenomic approaches from faecal samples, but the authors highlight that these resources are not yet in place and may be some time in development. The ITS nemabiome approach is currently being used - how does its performance limit its application? This may be a legacy of mine from reading papers that state their impact, but whilst I agree with the authors that we need more accurate, precise and practical tools for deciphering the mixture of cyathostomin species, I was not sure if they are suggesting the ITS nemabiome cannot be used? This requires some clarification.

The manuscript is generally very well written with only occasional lapses that require attention:
1. Occasional use of single sentence paragraphs - see Lines 348-349 and within lines 376-397 and elsewhere. Sentences should be brought together where possible.
2. Line 52 - Strongyles in italics
Line 101 and elsewhere - degenerated or degenerate (I am more familiar with the latter).
Line 120 DNA
Line 194 formatting needs attention
Line 330 remove 'in these' as not necessary as inferred.
Line 337 syntax

Figures look of good quality, however I think the legend to Figure 3 needs a little revision - can teh authors please define the 'different sample types'? I assume the erm 'pool' refers to adult parasites expelled post treatment?
Tables are clear but are inconsistent in terms of horizontal lines and vertical lines (included in some but not others). All tables need to be checked to make sure they are consistent and comply with journal guidelines.

Experimental design

The research question is well defined and is meaningful. The study lays out the importance of the work and the knowledge gap it is addressing.

In terms of rigour, the development of mock cyathostomin populations is robust, well described and well thought through. This, along with the use of multiple, different species compositions, and replicate studies using two different barcodes translates into high technical standards.

The methodology is described with detail and clarity.

Validity of the findings

The findings are well stated and linked to the original question, notwithstanding the comment above re articulating how the current use of existing ITS nemabiome may limit its use.

Underlying data are appropriate and all relevant material is provided and they are robust.

I am not in a position to extensively critique the detail of the analytic aspect of the study (e.g analytical pipeline and statistics), but overall the in depth description and attention to detail in many aspects of the experimental design are clear and should be commended.

Additional comments

I commend the authors for a very well conducted study on the importance of using mock cyathostomin communities to clarify the use of ITS-2 to detect species composition in complex infections. The inclusion of a second barcode, the CO1, was welcomed and thorough evaluation of its utility (or lack of) will be of considerable value within the field. The approach taken appears robust and well described with all relevant analyses and datasets well defined. It was a very well written paper that was enjoyable to read.

I have only one question. I agree that we need improved methods for accurately detecting cyathostomin species, however it was not clear to me if the authors are suggesting that the ITS nemabiome should not be used, due to its limitations. Is this the case? Given that the CO1 did not perform even as well as the ITS and comprehensive mitogenomes and nuclear genomes are likely going to take many years to develop, what do the authors propose should be done? Perhaps a sentence or two could be added to the discussion to make reference to this?

Reviewer 2 ·

Basic reporting

The use of English is appropriate and clear for most sections of the manuscript. However, word choice can be ambiguous and may be technically inappropriate at some points.

Key literature background and synthesis is lacking. Most background regrading the nemabiome is from two papers: Avramenko et al. 2015; Poissant et al. 2021. I agree that both of these papers are critical literature, but there is an overall lack of general ‘nemabiome’ literature. The nemabiome approach has been applied to sheep, zebras, deer, bison, etc.. Limited mention of these papers (and the suggestions and conclusions derived from them) result in a narrow scope of nemabiome background information. As such, readers of this paper not familiar with the nemabiome approach are likely to have a misrepresented understanding of the current strengths and weaknesses of the approach. As this manuscript suggests that metabarcoding will ‘unravel yet unknown facets of cyathostomin phenology’, providing evidence for its use across diverse study systems is necessary for support, since it has already been applied to a variety of studies.
It is true that mock communities haven’t been used for equine strongylid communities. However, similar work on equine strongyles has been conducted by Poissant et al. 2021. Poissant et al. 2021 did not ‘create’ mock communities, but conducted morphological identification on the same metabarcoded samples, resulting in a similar ‘outcome’ as the approach used by the current authors, albeit not for cyathostomins. In that regard, I believe authors should emphasize the differences for their approach and how it contrasts to previous equine work. Furthermore Avramenko et al. 2015 and Redman et al. 2019 created mock communities for cattle and ovine nemabiome communities previously. It would be inappropriate to discuss mock communities without the context of such experiments being performed on other study systems. Ultimately, there is a lack of synthesis of this manuscript’s results from the mock community experiments relative to previous literature results.
In respect to testing the differences between eggs, larvae, and adult worms, the authors are missing critical literature. Within larvae life stages there has been a few papers that test possible differences between L1 and L3 in non-equine systems. Borkowski et al. 2020 compared L1 and L3s for small ruminants (sheep and goats) while Redman et al. 2019 compared eggs, L1, and L3. Furthermore, there is evidence of pyrantel resistance in cyathostomins (Slocombe and de Gannes 2006; Nielsen et al. 2021; among others). If adults worms were compared to shed eggs/larvae, there needs to be recognition that the adult worms sampled may be biased. Brief mention of drug resistance is in the discussion, but with no interpretation of how this may contribute to biases in the current manuscript. Furthermore, variation in fecundity of adult worms should also be recognized as a possible bias when comparing eggs/L3 to adults. The authors’ conclusion that there was no differences in sample type on metabarcoding results suggest that some of these biases may not be significant for equine cyathostomin communities. Kuzmina et al. 2012 could be used as support for this result to suggest that cyathostomin species (many of which are included in this manuscript) have similar fecundity, thus the lack of bias.
Ultimately, insufficient background and literature use makes me cautious that the interpretations and conclusions from this manuscript are lacking.

The structure of the manuscript is broadly fine. However, there are instances where certain sentences should be removed or shifted to another section. For example, interpretations should not be present in figure captions, and justification for experiment should be in methods, not discussion. Some figures are misleading and may not clearly reflect the analyses. Figure captions are also lacking detail, resulting in further ambiguity of how the figures were made and what result/conclusion they are visualizing. Furthermore, there are errors in the text when describing figures, while other times, reference to figures in the text do not translate over easily.
I admit that I am not familiar with the data structure or many of the programs used in this study. Thus, I am unable to make significant judgments on many aspects of the data (particularly related to the COI barcode).

Hypotheses and the background literature grounding the hypotheses are unclear, nor are results explicitly linked to the hypothesis they address. The objectives of this manuscript are broad, with each objective having an associated hypothesis. Possibly due to the broad scope of the paper, I feel that the authors have focused on breadth instead of depth, and do not have clear resolutions to each of their objectives. This is particularly reflected in the discussion where there are limited interpretations of the variety of results obtained by the authors. Furthermore, the conclusions are lacking explicit connection to the objectives/hypothesis, and lead to a lack of cohesion from introduction to conclusion.

Experimental design

The research fits within the Aims and Scope of PeerJ.

Research questions are not well defined or synthesized. Unfortunately, there is not enough background literature, resulting in the identification of 'knowledge gaps' which may have already been addressed. This is of particular concern, as the authors are providing mostly ‘null’ results (i.e., metabarcoding is accurate, life stage doesn’t matter, new primer isn’t as good as established primer) for which some evidence is already available. I believe such null results are critical for the development and adoption of the nemabiome approach, however, I feel the authors need to explicitly mention that work similar to their manuscript is available within and beyond the equine literature.

Based on the analyses performed, I do not feel as though rigorous investigation was conducted. I am especially concerned about: 1) the reasoning behind using DNA to create communities instead of worms, 2) limited sample size, 3) using DNA from only one individual per species. There is no consideration for these concerns in experimental design (justification) or analyses/interpretation (bias). If these points are not explicitly addressed, I would be concerned about the depth of investigation.

There are no details on the reasoning or basis for many methodological decisions. Furthermore, there is limited detail in certain aspects (i.e., were fecal parasite samples standardized, if not, why?) that limit the ability to replicate. These details must be added, or the authors must refer to literature that they emulated the methods of.

Validity of the findings

Some of the objectives of the manuscript have been addressed in literature in different study systems. I believe that such literature must be added to provide context. As described in the previous paragraph regarding missing literature, I hope the authors specify literature to indicate the value of mock communities. Particularly for Avramenko et al. 2015, the use of mock communities allowed for bias corrections after sequencing. Similar approaches could be possible using data from Poissant et al. 2021 (not a mock community, but morphologically identified). I would suggest that authors acknowledge the previous uses of the nemabiome to emphasize what results are unique to their current manuscript. Addition of such literature would contribute to the validity of the findings, particularly from a comparative perspective.
Validity of the findings is also limited due to the lack of synthesis in the discussion. I would suggest the addition of more literature specific to biases inherent in the methodology. In its current form, discussion of biases regarding: differences between primers, the usage and availability of a sequence database, potential biases in experiment, etc. are lacking. Thus the conclusions of the manuscript do not necessarily have the appropriate context to be confident.

The data is available and the associated script and programs are mentioned. Unfortunately, I am concerned with the statistical soundness of this manuscript. Particularly due to the use of a single individual for all mock communities, the usage of DNA instead of individual worms for communities, and the lack of sample size. Crucially, the authors did some replication for their communities. However, these 'replicates' were constructed from the same source, thus are not independent. The authors must give clear and explicit reasoning behind these methods to reassure that the analyses are appropriate. Otherwise, I am worried that confident conclusions cannot be made.

Conclusions of the paper are somewhat linked to the original objective. However, I would suggest that the authors be more explicit with the connections. I would suggest the structure of the conclusion to be laid out in the same sequence as how the objectives were mentioned in the introduction.
Support for the conclusions is also lacking and limited by the scope of the experiments conducted. Some results also deserve more investigation to confirm if there are any outliers due to error. It needs to be more evident that the authors have conducted analyses with thorough investigation of their own data.

Additional comments

I am impressed by the scope of the paper, but caution the authors on reaching conclusions within the scope of the experiment.

I am particularly happy to see the application of the COI barcode, and believe that this is important science necessary to expand adoption of nemabiome metabarcoding. I am glad to see that the authors are intent on publishing the suboptimal results of the COI barcode, as these investigations are necessary for advancement of the field, even if they may not appear 'exciting'.

As the breadth of the manuscript is broad, I would suggest a restructuring of the manuscript to emphasize the differences between COI and ITS2 as its main contribution. I believe shifting the focus from 'mock communities' to 'testing barcodes (using mock communities)' is a more compelling story from the results.

Annotated reviews are not available for download in order to protect the identity of reviewers who chose to remain anonymous.

---

## Round 0.2 · Minor Revisions

Dear Dr. Courtot and colleagues:

Thanks for revising your manuscript. The reviewers are very satisfied with your revision (as am I). Great! However, there are a couple issues to still address and a few minor edits to make. Please address these ASAP so we may move towards acceptance of your work.

Best,

joe

Reviewer 2 ·

Basic reporting

The context established in the introduction is well written, and the pertinence of the literature cited is evident. The introduction is compelling, and is thorough in guiding the reader through the reasoning and conception of the study. The literature grounds the hypothesis being tested, and the narrowed focus enhances the clarity of the manuscript. I commend the authors on revising the manuscript, and I greatly appreciate the edits made in response to the referee comments. The manuscript was enjoyable to read.

Figures show marked improvement from the initial version and are clearer in presentation and relevance. Justification for methods used to construct figures are clear, and provide the necessary context for interpretation.

Experimental design

The research question is well defined, and it’s conception as a response to limitations of current nemabiome metabarcoding approaches is evident. The development and testing of a novel COI barcode using experimental communities is a significant and novel contribution, especially when compared with the ITS-2 barcode. The manuscript clearly articulates the reasoning behind the sequence of experiments and is clearly linked to the questions presented in the introduction. The inclusion of justification for certain methods is appreciated.

I would like to commend the authors in taking the initiative to conduct additional experiments. The inclusion of additional qPCR, though not changing the conclusions, enhance the robustness of the manuscript.

Just a couple points for clarification:
1. For the ITS-2 quality control and filtering, can the authors specify the max number of mismatches permitted when merging forward and reverse reads? I am not familiar with the architecture of the script presented and was unable to explicitly find the value. There are some reads lost following merging when looking at the table tracking reads/sample. However, I doubt the conclusions would be dependent on changing this parameter.
2. The figure caption for Figure 3 is improved by stating that unclassified reads were not visualized. For the mock communities, the % unclassified reads were presented in the results, but this was not the case for the ponies. I recognize the % unclassified is not an important consideration for this manuscript. However, I would recommend at least adding a range of % unclassified for the pony samples in the results or in the figure caption to further confirm that the implementation of the pipeline was robust (e.g., if the % of unclassified reads is high, I would be a bit suspicious of the quality of the sample or sequencing).

Validity of the findings

The findings presented in the manuscript are clear and conclusions are appropriate. Including more details in methods further support the validity of findings. The addition of more discussion on the limitations of this study is greatly appreciated, and I believe these additions have enhanced the conclusions. The thorough use of literature to compare between COI and ITS-2 in the discussion is well received, and further highlight the significant contributions of this manuscript.

The conclusions are well stated, and I believe the enhanced focus of the manuscript help maintain cohesion throughout. The conclusions does a great job of summarizing the results, while recognizing the limitations of the presented study.

One thing that may be relevant is inclusion of a sentence or two on the outlier samples (larvae results for ponies W734 and W748). It may be interesting to address these inconsistent communities, or if the authors can speak to these errors being of technical or biological origin. However, I recognize that it may be beyond the scope of this paper to suggest mechanistic reasons to why it was only the L3 samples that appeared odd, especially given the limited number of horses in the study.

Additional comments

Very minor edits/typo corrections are suggested for consistency sake, but these are purely typographical and are not associated with the science presented in the manuscript.
- consistency in usage of 'spp' (e.g., L54) vs 'spp.' (e.g., L56) vs 'sp'. (e.g., L562). ‘spp’ is the most commonly used, but figures and table use sp.
- consistency in using ‘µl’ vs ‘µL’ (e.g., L217)
- consistency in putting a space after volume ‘2µL’ vs ‘2 µL’ (e.g., L219)
- L177: ‘th’ to ‘the’
- L237: ‘Beads’ should not be capitalized
- L475: ‘TThe’ to ‘The’

---

## Round 0.3 · accepted · Accept

Dear Dr. Courtot and colleagues:

Thanks for once again resubmitting your manuscript to PeerJ. I now believe that your manuscript is suitable for publication. Congratulations! I look forward to seeing this work in print, and I anticipate it being an important resource for groups studying equine microbiomes and nematode parasites. Thanks again for choosing PeerJ to publish such important work.

Best,

-joe